# The Influence of Built-Environment Factors on Connectivity of Road Networks in Residential Areas: A Study Based on 204 Samples in Nanjing, China

**Yu Zhang** [1,*]**, Rui Wang** [2,*] **, Yue Wu** [1]**, Guanlong Chu** [1] **and Xiaomin Wu** [1]

[1] School of Architecture, Southeast University, Sipailou 2, Nanjing 210096, China
[2] Research Institute of Architecture, Southeast University, Sipailou 2, Nanjing 210096, China
**\*** Correspondence: 101010614@seu.edu.cn (Y.Z.); wr0624@seu.edu.cn (R.W.);
Tel.: +86-(25)-8379-5689 (Y.Z.); +86-(25)-8379-3271 (R.W.)

**Abstract:** Over the past decades, China has built a large number of superblocks and gated residential areas, which have contributed to increased congestion on urban road traffic. Although some studies have found a correlation between residential environments and travel convenience, there is little quantitative evidence to explain which factors influence the connectivity of road networks. This paper aims to clarify the indicators of built environments in Chinese residential areas that affect travel accessibility and their correlation proportions through a more quantitative method and to provide a basis for planning and design decisions. To this end, this study took 204 residential areas in the three districts of Nanjing, China as a sample and calculated 16 built-environment indicators. The path distance ($D$) and pedestrian route directness ($PRD$) from the center of these residential areas to the nearest urban intersection were measured by GIS as outcome variables. First, correlation analysis was used to screen for significant variables, and multiple linear regression models were used to examine the significant influencing factors of $D$ and $PRD$. Then, a binary logistic regression analysis was performed to provide a model that could determine whether the built environment of the residential area met the requirements for convenient travel. The results revealed that the length of the long side, the area size, and the total road length of the residential area were significantly related to $D$, and the number of entrances and exits, the intersection density, and the X ratio were significantly related to $PRD$. The indicators that were positively correlated with $D$ were the size, network complexity, and the boundary shape of the residential area according to the degree of correlation. Meanwhile, other indicators such as the density and connectivity of the residential road network were negatively correlated with $D$.

**Keywords:** residential area; built environment; connectivity; road network

## 1. Introduction

Residential areas are the basic units of urban composition. The design pattern of the settlement and the choice of development modes directly affect the development of the whole city. During the end of the 20th century and the beginning of the 21st century, China's urban construction often adopted a mode of "development first, planning later", and the residential area adopted the development mode of a "Residential district" model. Residential areas have expanded to the periphery of the city, generating a large number of suburban residences, and the urban suburbs have gradually become a large-scale, introverted super-block area [1,2]. The construction mode of the super-block area and the residential community is dismembering the overall road network of the city. The internal road network of the residential community and urban traffic are independent factors without any overall consideration. The arising of such suburban super blocks is accompanied by the rapid growth of the number of private cars, which has aggravated the trend of urban sprawl. The sprawling has in turn created a huge impact on traditional

high-density and mixed-use urban forms, resulting in traffic congestion, a sharp increase in commuter travel distances, and high carbon emissions from traffic [3–6].

Numerous studies have shown that the built environment of a residential area and urban traffic are significantly related [7–10]. Cervero (1997) attributed the factors affecting residents' traffic behavior to three important dimensions (3Ds), namely, density, diversity, and design [11]. Brown et al.'s (2009) research indicated that the higher the mixedness of a community, the more likely the residents are to choose to walk when they go out [12]. Glaeser (2008) used the National Household Energy Data to analyze the traffic travel of residents in 66 cities in the United States and considered density to be an important indicator reflecting commuting distance [13]. Brownstone (2009) held that density affects the vehicle type selection preference when residents go out [14]. Ewing (2010) used Meta's analytical method for statistical analysis to find that the mileage of the vehicle is most relevant to the reachability of the destination [15].

Most of the previous research has been carried out at the level of the city and the block, and it was less deeply attuned to the organization of the physical space form within the residential area. More than 80% of the residential areas in China are enclosed districts, which are connected to the outside city roads with a small number of entrances or exits. Roads inside a residential area cannot be used by residents outside the area. As such, the internal road network of the residential area does not participate in the regional "road connectivity" and "road network density". It has little influence on smaller residential communities, but a significant impact on larger ones. One of the key issues to be solved in this paper is how to reflect the influence of the internal road network configuration on the "connectivity" of the road and integrate it into the urban road network system for measurement and control in China. To achieve this aim, 204 representative residential areas with significant differences in built environment, development background, and location in the three districts of Nanjing, China were selected for research, and the built-environment indicators and road network connectivity in the residential areas were calculated quantitatively, which can establish a model to be tested.

## 2. Literature Review

Connectivity refers to the number of directions and paths to a destination, and it is an indicator of measuring how easy it is to travel between locations [16]. High connectivity is the goal of any transportation network. Good road connectivity can reduce travel distance and time, multiply the choice of travel routes, reduce unnecessary bypass distance, and increase accessibility. Road connectivity is meant to "improve accessibility and reduce traffic barriers" [17]. The factors that determine road connectivity often involve efficiency, distance, and the diversity of options, and the attribute of road connectivity depends on the trade-offs among these three factors. For pedestrian traffic, a well-connected road network is equipped with many short links, numerous notes, and fewer culs-de-sac. As the travel distance decreases and the path selectivity increases, destinations are more easily connected directly, all of which contributes to creating a more accessible and flexible transportation system. On the other hand, for cars, a higher traffic speed often requires less connectivity by reducing intersections and restricting pedestrian crossings. Thus, the connectivity of car and pedestrian traffic is sometimes contradictory.

### 2.1. Street Pattern and Connectivity ("Grid" or "Tree" Road Network)

Some studies believe that a "grid" road network is provided with better road connectivity and control flexibility, whereas a "tree" road network and those with more culs-de-sac are the opposite. A "branched" road network should be avoided as much as possible. Grid streets characterized by short street segments and frequent intersections are provided with higher connectivity [18]. Since the distribution of center values in the street pattern (node/link) is more uniform, and it is not easy to be interrupted at a certain node due to a malfunction, the grid street is more adaptable in an emergency situation [19,20]. David Walters used the "Connectivity Index" to assess the connectivity of urban streets, pointing

out that the connectivity index of a perfect grid road is 1.4 [21]. Other calculation methods also prove that the grid road network is of higher connectivity [22].

However, Stephen Marshall (2005) argued that this is not always the case, and the tree pattern itself is not undesirable [23]. The hierarchical structure (the relationship between tree roots, branches, and twigs) found that the tree pattern follows a power-law distribution. This hierarchical structure provides the possibility of a semi-private public space, and it is also recommended to use a "leaf-shaped road network structure" with strong connectivity in the residential area.

Other controversies have shown that the high connectivity of grid road networks can also lead to shorter-range car travel and increase congestion and greenhouse gas emissions [24,25].

### 2.2. Road Network Density and Road Connectivity

A road network simulation analysis model established by Chinese scholars shows that the reliability of the level of road network service increases with the rise in road network density, and traffic efficiency tends to go up first and then fall off along with the growth of road network density [26,27]. In theory, there is an optimal road network density. Wu Shanglin (2016) believes that the low-carbon-oriented block scale should be controlled in the range of 100~200 m, and the total density of all types of road networks ought to reach 10 km/km$^2$~16 km/km$^2$ [28]. In China, the density of urban road networks is much smaller than that of developed countries in the West due to the large number of internal roads within the residential area, which are not included in urban roads [29]. Considering car traffic, one intersection per 200 m is too dense, whereas one intersection per 800–1000 m is unfavorable for residents' travel. It is suitable for both pedestrians and vehicles to have an intersection every 300–800 m. It is believed that this conclusion is worth discussing. Other factors, such as road width [30], road grade [31], and so on, all have an effect on low-carbon transportation.

### 2.3. Relevant Study on Connectivity Measurement

Means of measuring connectivity indicators differ among different scholars. Hess (1997) proposed to evaluate the connectivity of streets in different urban areas according to the square meters of the qualified Pedestrian Route Directness (*PRD*) area [32]. Randall and Baetz (2001) used *PRD* and distance to measure connectivity levels and established evaluation criteria for walking paths between departure points and destinations [33]. Saelens (2003) pointed out that high connectivity can lead to more path choices, which is beneficial to shorten travel distances and increase travel frequency [34]. Jennifer Dill (2004) believes that most travel behaviors are purposeful [35]. A highly connected road network can help to save time spent on the road, and therefore, this indicator is applicable to any traffic network. Frank (2004) analyzed the influence of intersection elements on connectivity [36]. It was found that the higher the density of intersections, the higher the frequency of residents' travel behavior will be within a radius of 1 km. Nelson (2006) used coefficients such as intersection density, road accessibility, road network complexity, and other factors as an index to describe connectivity [37]. He selected the correlation analysis of the connectivity within a 3 km radius and found that connectivity was positively correlated with the walking level. Owen (2007) analyzed the influence of different types of road networks on the accessibility of travel destinations from the perspective of a road network mode [38]. He believed that the "directness" and "selectivity" of the route are the evaluation criteria of connectivity and arrived at the conclusion that the connectivity of a grid road network is high. In practice, the city of Portland selected 1.5 as a standard for street directness and considered travel routes with a directness greater than 1.6 to be cases of poor connectivity [32]. The measurement method of road "connectivity" has been explored and is changing all the time in the architectural field. The most widely used method is the Pedestrian Route Directness (*PRD*), and this method is followed in this paper.

## 3. Analysis Method and Framework

This paper selects 204 residential areas in Jianye District, Jiangning District, and Pukou District of Nanjing as examples to investigate the relationship between the residential environment and road connectivity. The selected three urban areas are all located on the outer periphery of the main city of Nanjing. Each center of the district is about 10 km away from the main city, which belongs to the sub-center of the city. The area of Jianye District is about 9.7 square kilometers, in which there are 76 residential area samples; the area of Jiangning District is about 9.8 square kilometers, in which there are 61 residential area samples; and the area of Pukou District area is about 12 square kilometers, in which there are 67 residential area samples.

### 3.1. Research Object and Area

The characteristics of the selected 204 residential areas are: (1) significant differences in the built environment of residential areas, such as size, road scale, mixability of land use, population density, and surrounding facilities of the three regions (Table 1); (2) for the development of different years, the planning and design concepts of residential areas are quite different; (3) representative of developed areas includes factors such as similar building capacity-controlling indicators, including volume ratio, building density, building height limits, and other indicators.

**Table 1.** Characteristics of 204 residential areas investigated in 3 administrative districts.

|  | Jianye District | Jiangning District | Pukou District |
|---|---|---|---|
| Completion time | 2006–2016 | 1998–2010 | 2010–2016 |
| Planning mode | Exploitation before planning | Planning before exploitation | Old renewal |
| Mode | Blocks | Neighborhood units | Housing estates |
| Scale | Medium plot | Oversized and large plot | Oversized and large plot |
| Residential type | High-rise and a few multi-story buildings | Multi-story and a few high-rise buildings | Multi-story and a few high-rise buildings |
| Number of residential areas | 76 | 61 | 67 |

### 3.2. Data Acquisition Method

Residential environment data acquisition was divided into two parts: artificial field measurement and online map extraction. The field measurement was mainly carried out by the researcher through taking photos and field exploration and verification of the road network form of the residential sample. The online map extraction was achieved by downloading satellite images of the road network in residential areas through relevant websites and big data sources and drawing a road network base map in ArcGIS software (version 10.2). The road network base map generated by the online map was supplemented and corrected through the second field of research.

The main methods for obtaining the built-environment and connectivity indicators in residential areas are:

- Using the API (application programming interface) interface of a public map website and the national geographic database to obtain the road network data of the research area and the location coordinates of the center point of the residential area sample and screening samples of research significance.
- Using a satellite map as the base map in ArcGIS software to draw the urban road network map of the research area and establishing a database for the sample of the residential area, then inputting the basic information and coordinate positions.
- Drawing a road network map of the residential area containing points (such as intersections, entrances and exits) and lines (residential roads) for each sample, then connecting it spatially with the urban road network map in ArcGIS software (Figure 1); establishing a network analysis dataset in ArcGIS software and using the spatial analysis tool to count and calculate the road network shape indicators of the residential area sample. For example, by means of spatially analyzing the number of line elements connected to each node element, we can distinguish and count the number of culs-de-

sac, T-junctions, and X-junctions; finally, summarizing and graphical representation are performed in Excel.

- In the ArcGIS software, the path analysis tool is used to calculate the linear distance and the shortest path distance (*D*) from the geometric central point to the adjacent city intersection of each residential area sample, and the *PRD* is calculated.

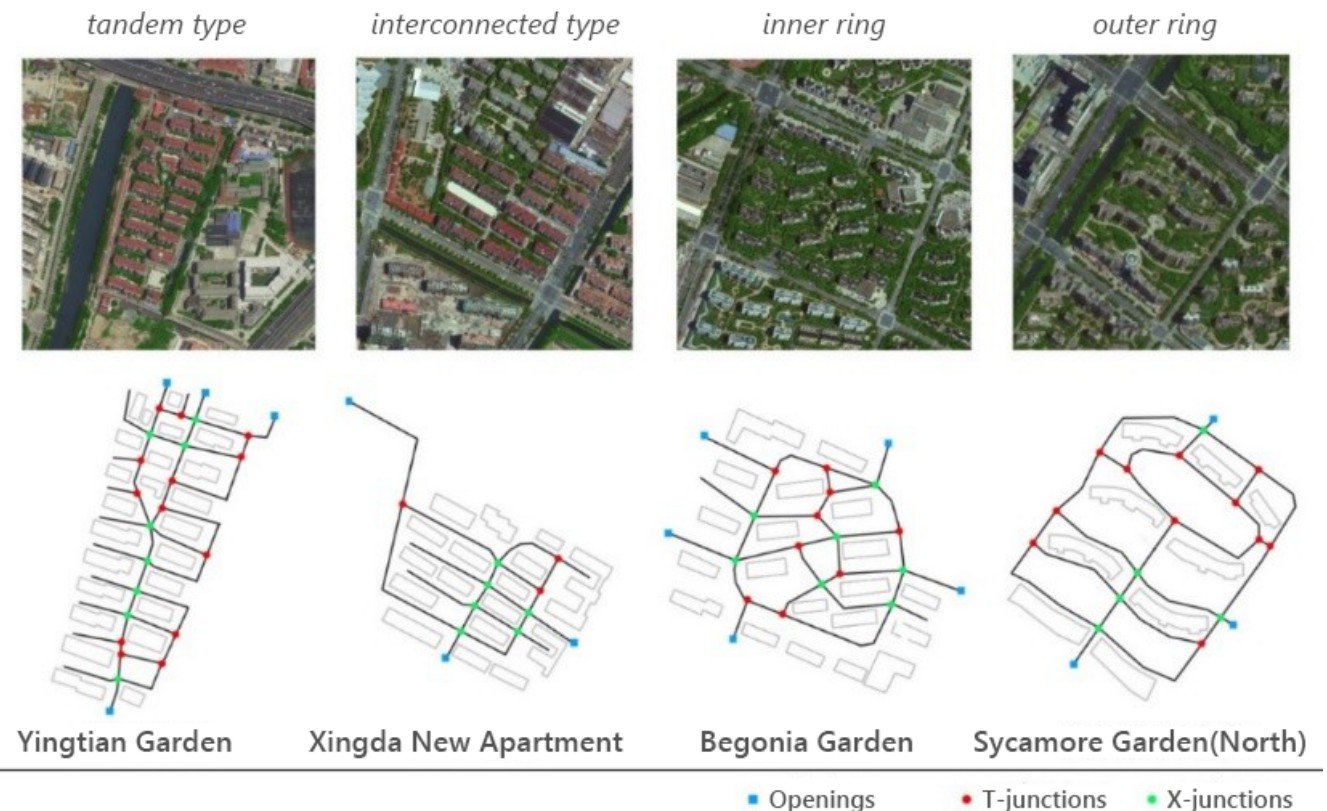

**Figure 1.** Schematic plot of road network extraction in residential areas.

### 3.3. Analysis Framework

As shown in the framework, the data of the built-environment indicators and road connectivity indicators of the residential area are measured separately (Figure 2, Table 2). Built-environment indicators include the number of entrances and exits *N1*, the length of the long side of the residential area *N2*, the length of the short side of the residential area *N3*, the ratio of the long and short sides *N4*, the area size of the residential area *N5*, the total road length *N6*, road network density *N7*, the number of nodes *N8*, the number of links *N9*, intersection density *N10*, the number of X-junctions *N11*, the number of T-junctions *N12*, the X ratio *N13*, cell road *N14*, culs-de-sac *N15*, and cell ratio *N16*—16 items in total. Road connectivity indicators include path distance (*D*) and pedestrian route directness (*PRD*). An analytical prediction model is built between each of the indicators of the built environment of the residential area (*N*), the path distance (*D*), and pedestrian route directness (*PRD*). The steps to establish an analytical prediction model are as follows: (1) In the SPSS software, Pearson correlation analysis is used to calculate the linear correlation between the road network shape index and the path distance (*D*) as well as the pedestrian route directness (*PRD*). (2) Using the SPSS software, multiple linear regression analysis is used to build a comprehensive calculation model for path distance (*D*) and pedestrian route directness (*PRD*). (3) Using SPSS software, binary logistic regression analysis is used to establish an application judgement model for whether road connectivity is acceptable and to apply the model for a connectivity measurement test.

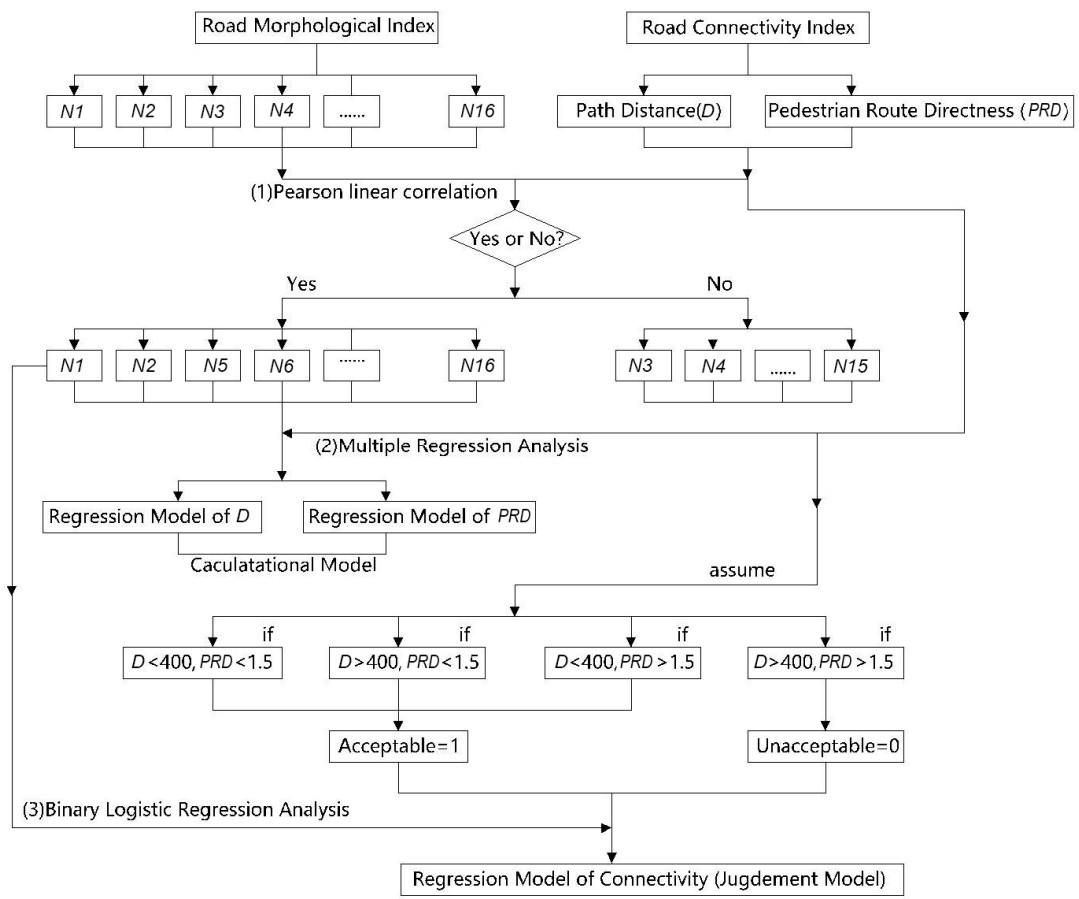

**Figure 2.** Framework plot of analysis process and method.

**Table 2.** Variables for built environment indicators and road connectivity indicators.

| Variable | Indicators | Variable | Indicators |
|----------|-----------|----------|-----------|
| N1 | number of entrances and exits | N10 | intersection density |
| N2 | length of the long side of the residential area | N11 | number of X-junctions |
| N3 | length of the short side of the residential area | N12 | number of T-junctions |
| N4 | ratio of the long and short sides | N13 | X ratio |
| N5 | area size of the residential area | N14 | cell road |
| N6 | total road length | N15 | cul-de-sac |
| N7 | road network density | N16 | cell ratio |
| N8 | number of nodes | D | path distance indicator |
| N9 | number of links | PRD | pedestrian route directness |

In this table,

$$N5 = N2 * N3 \tag{1}$$

$$N7 = \frac{N6}{N5} \tag{2}$$

$$N8 = N1 + N11 + N12 + N15 \tag{3}$$

## 4. Analysis and Discussion

### 4.1. Quantitative Calculation of Indicators of Built Environment of Residential Areas

Sixteen indicators (*N*) of the built environment of 204 residential areas in Nanjing were selected for quantitative measurement and calculation. Due to space limitations of the article, six important items among the 16 indicators were selected for explanation.

### 4.1.1. Number of Entrances and Exits

The data show that the distribution of the number of entrances and exits of residential areas in these three districts are close. Jianye and Pukou District mainly have about 2–4, and Jiangning District mainly has about 2–5. In Jianye, 32.89% of the residential areas have two entrances and exits, whereas the number is 26.26% in Jiangning and 34.44% in Pukou. In Jianye, 27.63% of the residential areas have three entrances and exits, whereas it is 31.15% in Jiangning and 26.87% in Pukou. Due to the large size of the residential area in Pukou District, 16.42% of the residential areas have four entrances and exits. The average and median number of entrances and exits in the three districts are close to three, indicating that regardless of the scale of the residential area, two or three entrances and exits account for the largest number (Figure 3 and Table 3).

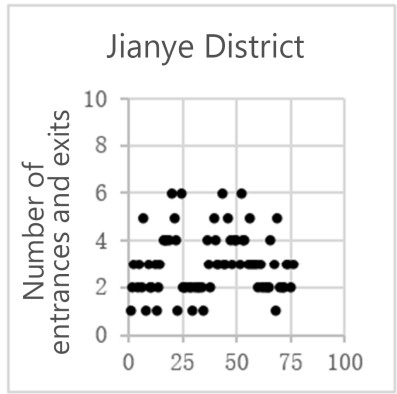 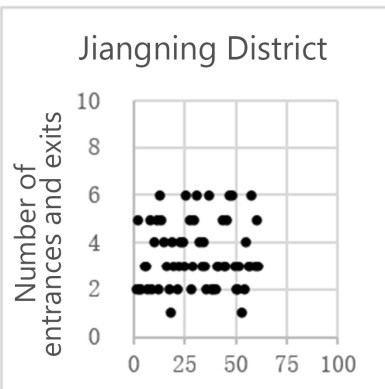 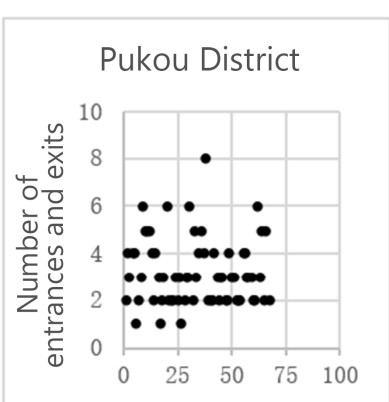

**Figure 3.** Numbers of entrances and exits of 204 residential areas in Jiangning, Jianye, and Pukou District.

**Table 3.** Numbers of entrances and exits of 204 residential areas in Jiangning, Jianye, and Pukou District.

|  | Number of Residential Areas | Minimum(M) | Maximum(X) | Average(E) | SD |
|---|---|---|---|---|---|
| Jianye District | 76 | 1 | 6 | 2.97 | 1.296 |
| Jiangning District | 61 | 1 | 6 | 3.44 | 1.409 |
| Pukou District | 67 | 1 | 8 | 3.19 | 1.417 |
| Total | 204 | 1 | 8 | 3.19 | 1.377 |

### 4.1.2. Scale of Residential Areas

Overall, the size of the residential areas in the three districts is generally large. The residential areas in Jianye District are primarily about 4–9 hectares, with an average of 7 hectares. They are mainly small- and medium-sized, without large residential areas. The residential areas in Jiangning District are primarily about 7–18 hectares, with an average of 15 hectares. Among them are mainly large-scale areas, whereas residential areas larger than 40 hectares account for 3.38%, and the largest one is about 60 hectares. The residential areas in Pukou District are primarily about 6–16 hectares, with an average of 13 hectares, more than 1.7 times that of Jianye District. They are mainly medium-sized, with a maximum living area of about 46 hectares (Figure 4 and Table 4). The proportion of super-large-scale residential areas of about 40 hectares is 5.97%. The farther away from the main city, the larger the size of the residential area. This is consistent with the incomplete early controlled detailed planning, as well as phenomena existing in the regions that are far from the main city, such as uncontrolled exploitation, low land prices, and large amounts of land acquisition by enterprisers.

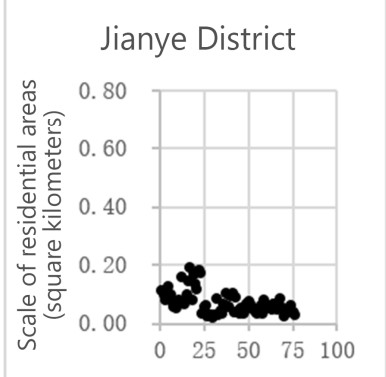
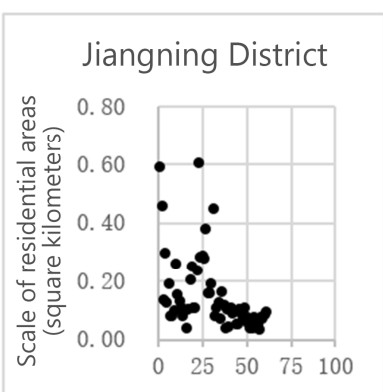
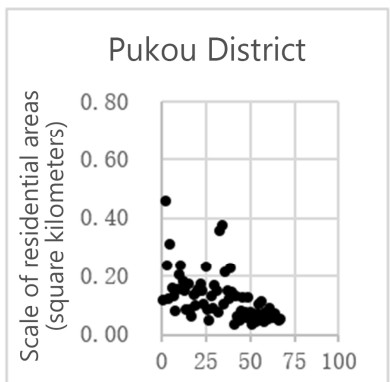

**Figure 4.** Area size of 204 residential areas in Jiangning, Jianye, and Pukou District.

**Table 4.** Area size of 204 residential areas in Jiangning, Jianye, and Pukou District.

|  | Number of Residential Areas | Minimum (M) | Maximum (X) | Average(E) | Standard Deviation |
|---|---|---|---|---|---|
| Jianye District | 76 | 0.0223 | 0.1893 | 0.070612 | 0.038861 |
| Jiangning District | 61 | 0.0325 | 0.6051 | 0.149537 | 0.127645 |
| Pukou District | 67 | 0.033 | 0.4611 | 0.127061 | 0.083733 |
| Total | 204 | 0.0223 | 0.6051 | 0.112752 | 0.093767 |

### 4.1.3. Road Network Density

Road network density refers to the length of the road on a unit area within the residential area, which can reflect the proportion of space occupied by the road network. The density of the road network in the Jianye residential area is mainly about 20–32 km/km$^2$, with an average of 26 km/km$^2$. The density of the road network in the Jiangning residential area is mainly about 20–33 km/km$^2$, with an average of 27 km/km$^2$. The density of the road network of the residential area in these two districts is close, whereas the density value of the road network in Pukou is mainly about 16–25 km/km$^2$, with an average of 21 km/km$^2$, which is lower than the first two districts. The reason for this may be that Pukou district is smaller and farther away from the main city than the other two districts, and the planning structure is looser, so that the road network density is not high. Overall, the average internal road network density in the residential area is as high as 27 km/km$^2$, and those networks exceeding 30 km/km$^2$ account for 29.17% of the total, which is significantly higher than the road network density of 8 km/km$^2$, but since most of the residential areas are large and closed, the internal roads in the residential areas have not been fully used (Figure 5 and Table 5).

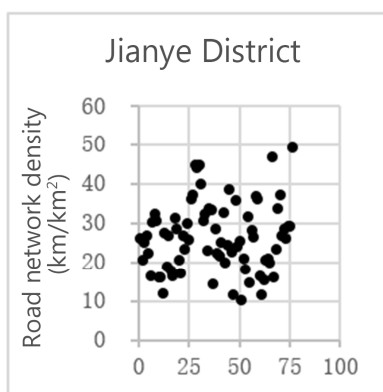
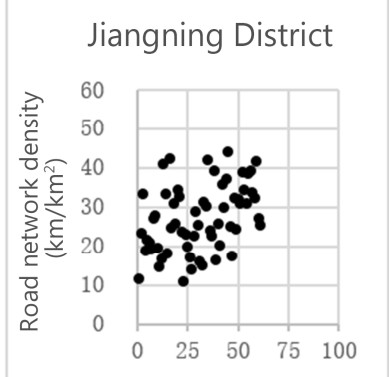
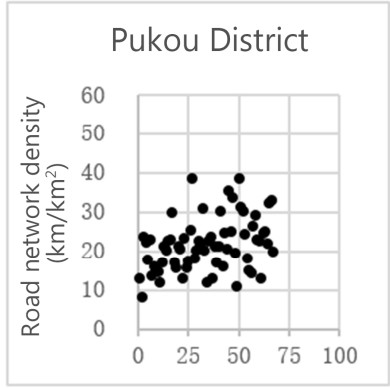

**Figure 5.** Road network density of 204 residential areas in Jiangning, Jianye, and Pukou District.

**Table 5.** Road network density of 204 residential areas in Jiangning, Jianye, and Pukou District.

| | Number of Residential Areas | Minimum(M) | Maximum(X) | Average(E) | Standard Deviation |
|---|---|---|---|---|---|
| Jianye District | 76 | 10 | 49 | 26.42 | 8.849 |
| Jiangning District | 61 | 11 | 44 | 27.2 | 8.54 |
| Pukou District | 67 | 8 | 39 | 21.59 | 6.779 |
| Total | 204 | 8 | 49 | 25.07 | 8.454 |

### 4.1.4. Combined Plot

The intersection forms of the road network in residential areas are generally X-junctions and T-junctions. The X-junction usually appears at the intersection of roads of the same level. The T-junction usually appears at the connecting point of the main and secondary road. Road networks dominated by X-junctions usually present a strong grid feature, and road networks dominated by T-junctions often present a tree-like feature. According to the combined plot proposed by Marshall [23], a two-dimensional coordinate space of "T ratio–X ratio" and "cell ratio–cul ratio" can be constructed. This quantitative measurement method helps to describe the degree of how "griddy" or "tree-like" the road network appears, and the intersections and road forms are classified and compared (Figure 6). In the figure, the left, right, upper, and lower coordinate axes respectively represent the T ratio, X ratio, cell ratio, and cul ratio, so the four corner points respectively represent four pure "configuration" forms, where the upper right is a "pure T grid", the upper left is a "pure T tree", the lower right is a "pure X grid" and the lower left is a "pure X tree".

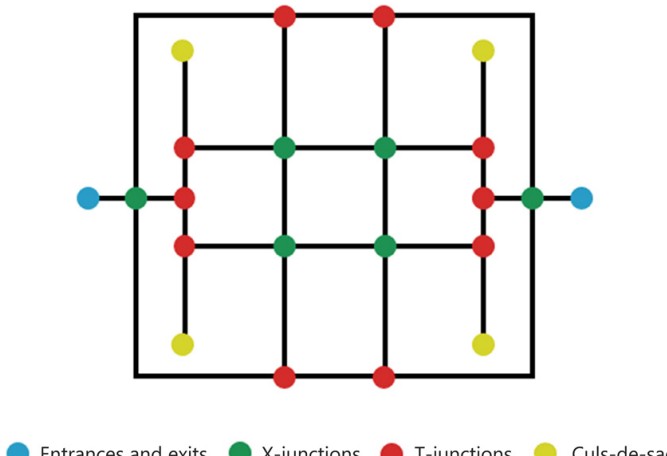

● Entrances and exits  ● X-junctions  ● T-junctions  ● Culs-de-sac

**Figure 6.** Schematic plot of Link-node.

According to the projection position of the four sets of data—the T ratio, X ratio, cell ratio, and cul ratio—of the residential area road network sample in different districts in the coordinate space, the primary road network type of the residential areas in the very district can be judged. Most of the road network type of residential areas in Jianye District are distributed between "pure T tree" and "pure X tree" and are partial to an "X tree" form, which can be regarded as a "variant grid" road network; Jiangning District and Pukou District are distributed above the coordinate area, and most of the residential areas present the characteristics of "T tree", that is, road network characteristics based on a "tree shape" (Figures 7 and 8).

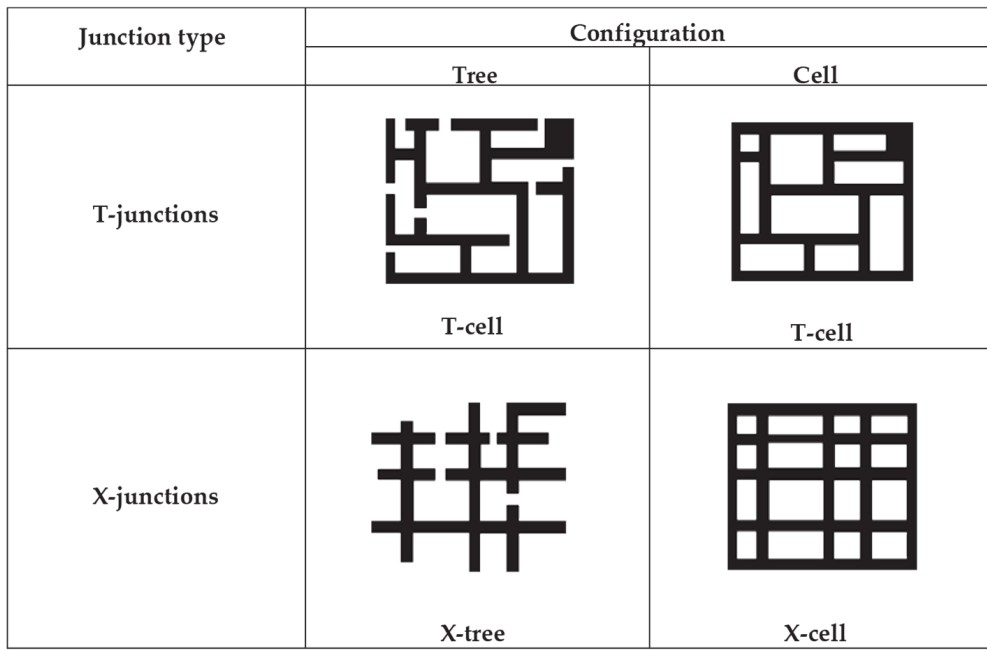

| Junction type | Configuration | |
| --- | --- | --- |
| | Tree | Cell |
| T-junctions | T-cell | T-cell |
| X-junctions | X-tree | X-cell |

**Figure 7.** Intersection form of the road network.

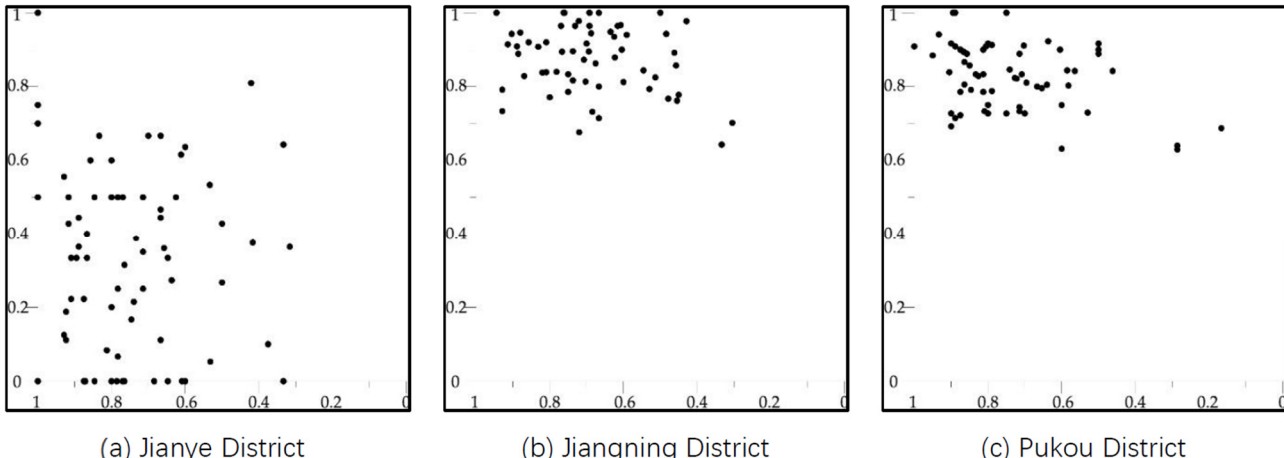

(a) Jianye District　　　　　(b) Jiangning District　　　　　(c) Pukou District

**Figure 8.** Combined plot of 204 residential areas in Jiangning, Jianye, and Pukou District.

### 4.1.5. Node Gram

The analysis method of the node gram is also derived from Marshall's research on the configuration of the road network. It mainly analyzes the relationship between T cross nodes, X cross nodes, and culs-de-sac (Figure 8), where the X-axis represents the proportion of the T-junction, the Y axis represents the proportion of the X-junction, and the Z axis represents the proportion of the cul-de-sac. These configurations are projected in a triangle chart, each point representing a road network sample. Thus, the proportion of different types of nodes can be distinguished according to the projection position of the T node, the X node, and the cul-de-sac in the chart. Points distributed on the left oblique side are the road network without an X-junction, and the right oblique side represents the road network without the cul-de-sac (Figure 9).

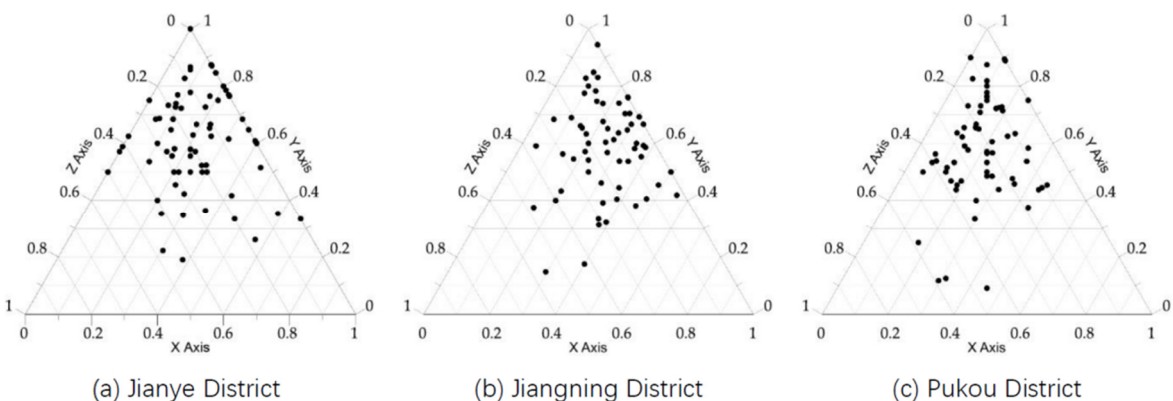

**Figure 9.** Node gram of the residential areas in Jiangning, Jianye, and Pukou District.

It can be seen from the graph that there are points of the samples in Jianye District distributed in both the middle and oblique sides of the triangle, indicating that the type of the road network in Jianye District is various and with an even amount. There are few points on the oblique side in Jiangning District and Pukou District compared with Jianye district, indicating that there are few extreme samples that have no X-junction or cul-de-sac. The node gram of Jiangning District shows that there are more residential areas of fewer culs-de-sac.

### 4.1.6. Link–Node Ratio

The node represents the set of all the connected endpoints in the road network. The more nodes, the more branches the road network will have. For a car route in a residential area, increasing the number of nodes can slow down the speed to some extent, but for a pedestrian route, the increase in the number of nodes is helpful for providing more path selection. In the residential road network, the opening of the link to the city road is also included.

The number of nodes in the residential area of Jianye District is mainly concentrated within 17–30, with an average of 25. Jiangning District is mainly concentrated within 28–53 nodes, with an average of 44, the most in the three regions. Pukou District is mainly concentrated within 17–35 nodes, with an average of 27. In contrast, the number of nodes in the residential area of Jiangning District is the highest, indicating that there are more intersections of different streamlines in the settlement, which is caused by the fact that the internal factors of the large-scale settlement are richer (Figure 10 and Table 6). On the other hand, the layout of elements, such as the architectural landscape, is related.

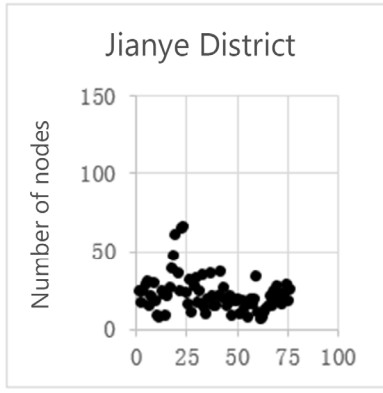
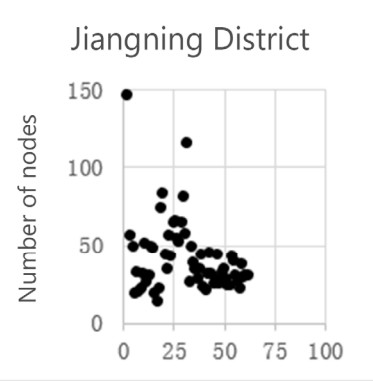
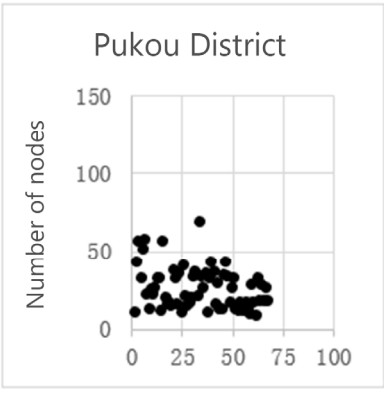

**Figure 10.** The number of nodes of the residential areas in Jiangning, Jianye, and Pukou District.

**Table 6.** The number of nodes of the residential areas in Jiangning, Jianye, and Pukou District.

|  | Number of Residential Areas | Minimum(M) | Maximum(X) | Average(E) | Standard Deviation |
|---|---|---|---|---|---|
| Jianye District | 76 | 8 | 67 | 23.17 | 11.769 |
| Jiangning District | 61 | 15 | 152 | 44.25 | 26.99 |
| Pukou District | 67 | 10 | 70 | 27.12 | 13.292 |
| Total | 204 | 8 | 152 | 30.77 | 20.113 |

### 4.2. Quantitative Calculation of Road Network "Connectivity" Index

The road network form and its connectivity play a decisive role in urban traffic improvement. The most critical issue is the scientific nature of quantitative analysis of the road network form. The difficulty here is the measurement and research of the "road connectivity". For the study of road "connectivity", there are many methods, such as the conventional traffic grid analysis method, space syntax, the path structure method, the link–node method, etc. Path distance (*D*) and pedestrian route directness (*PRD*) are used in this paper to measure the connectivity of roads in the blocks.

#### 4.2.1. Path Distance (*D*)

The path distance is calculated as the shortest distance from the center of the residential area to the road intersection outside. The path distance (*D*) in this paper is the average distance from the center point of a residential area to the intersections (a/b/c/d) on the four closest urban roads:

$$D = Average(Da + Db + Dc + Dd) \tag{4}$$

Connectivity is usually an indicator used to measure the car traffic efficiency, reflecting the car capacity of the road. When measuring the pedestrian traffic connectivity, it is necessary to add the limit of the maximum walking distance and to consider that the willingness toward walking decreases as walking distance increases. Most scholars from China believe that a 300 m radius is a comfortable walking distance [28,39]. Domestic scholars in China agree that 400~500 m is more suitable [29,30,40,41]. A total of 400 m around the residential area is used as the limit of walking distance in this paper.

According to the calculation results of the 204 residential areas, the distance between the residential areas of Jianye District and the intersections of the city is mainly distributed in the range of 278–443 m, and the average walking distance is 363 m. This distance is suitable for general walking. However, in Pukou District and Jiangning District, the path distance of the residential area is mainly distributed in the range of 300–600 m, with an average distance of 450 m. The path distance of more than 50% of the residential areas in these two districts is longer than 450 m, which is much higher than the reasonable expectation of daily walking (Figure 11 and Table 7). The residential areas of Pukou and Jiangning District are not suitable for residents to walk.

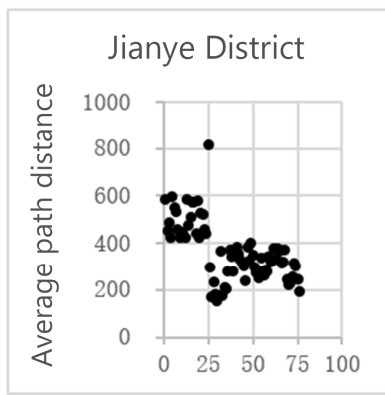
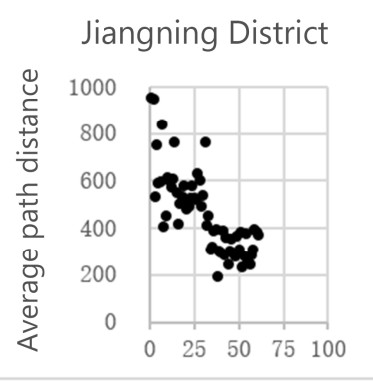
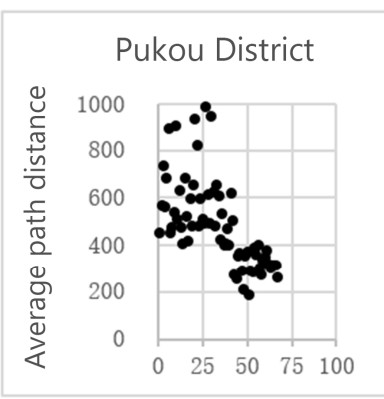

**Figure 11.** Average path distance of 204 residential areas in Jiangning, Jianye, and Pukou District.

**Table 7.** Number and percentage of residential areas with different levels of path distance.

| Path Distance | Jianye District | | Jiangning District | | Pukou District | |
|---|---|---|---|---|---|---|
| <200 | 6 | 7.89% | 1 | 1.64% | 1 | 1.49% |
| 200 ≤ 400 | 45 | 59.21% | 27 | 44.26% | 24 | 35.82% |
| 400 ≤ 600 | 24 | 31.58% | 22 | 36.07% | 26 | 38.81% |
| 600 ≤ 800 | 0 | 0.00% | 8 | 13.11% | 10 | 14.93% |
| 800 ≤ 1000 | 1 | 1.32% | 3 | 4.92% | 6 | 8.96% |
| Total | 76 | 100.00% | 61 | 100.00% | 67 | 100.00% |

### 4.2.2. Pedestrian Route Directness (*PRD*)

Pedestrian Route Directness (*PRD*) (Figure 12) indicates the ratio of the actual walking distance ($D_{reality}$) and the straight-line distance ($D_{beeline}$) between the starting point and the destination (Randall, Baetz, 2001).

$$PRD = \frac{D_{reality}}{D_{beeline}} \tag{5}$$

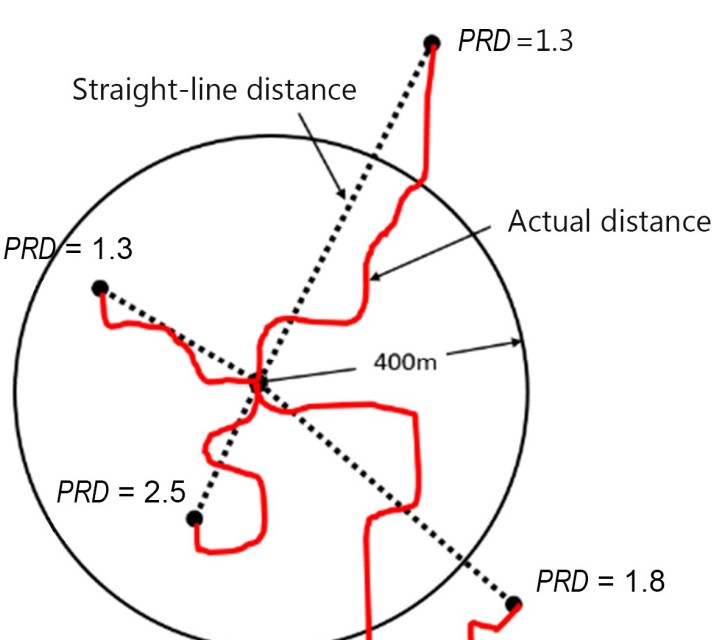

**Figure 12.** Pedestrian Route Directness (*PRD*).

In the ArcGIS software, the path analysis tool is used to calculate the linear distance and the shortest path distance of each residential sample from the geometric central point to the adjacent city intersection, and the *PRD* coefficient is calculated (Figure 13 and Table 8). From the results of data distribution, the average *PRD* value of residential areas in Jiangning District is 1.69, and most of the samples are below the average level. The average value of 1.77 in Jianye District is better than the value of 1.84 in Pukou District, but the data are mainly distributed in the range of 1.53–1.95. Compared with the data of 1.41–1.98 in Pukou District, there are fewer samples whose directness is acceptable. In conclusion, the directness level of the road network in residential areas in the three districts is not desirable, as the average values of *PRD* are all greater than 1.6.

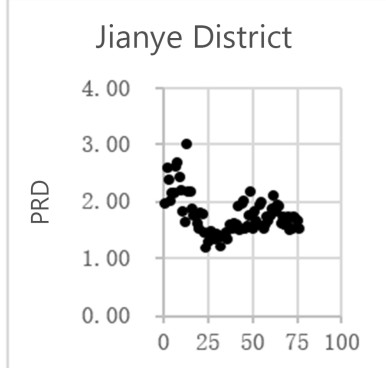
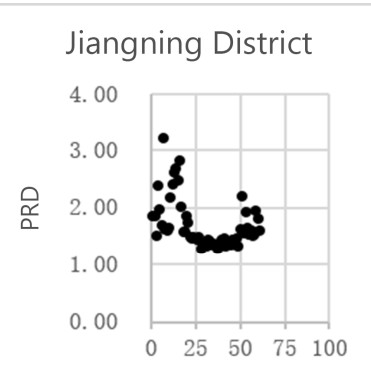
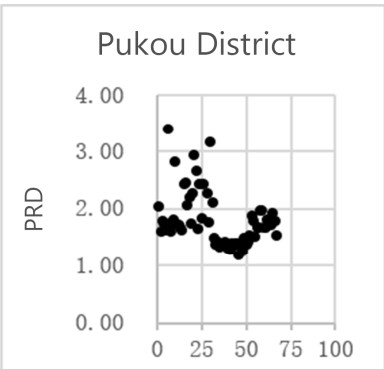

**Figure 13.** Average *PRD* of 204 residential areas in Jiangning, Jianye, and Pukou District.

**Table 8.** Number and percentage of residential areas with different levels of *PRD*.

| PRD | Jianye District | | Jiangning District | | Pukou District | |
|---|---|---|---|---|---|---|
| <1.2 | 1 | 1.32% | 0 | 0.00% | 1 | 1.49% |
| 1.2 ≤ 1.4 | 6 | 7.89% | 16 | 26.23% | 11 | 16.42% |
| 1.4 ≤ 1.6 | 21 | 27.63% | 19 | 31.15% | 11 | 16.42% |
| 1.6 ≤ 1.8 | 20 | 26.32% | 9 | 14.75% | 21 | 31.34% |
| 1.8 ≤ 2.0 | 13 | 17.11% | 7 | 11.48% | 7 | 10.45% |
| 2.0 ≤ 3.0 | 14 | 18.42% | 9 | 14.75% | 13 | 19.40% |
| 3.0 ≤ 5.0 | 1 | 1.32% | 1 | 1.64% | 3 | 4.48% |
| Total | 76 | 100.00% | 61 | 100.00% | 67 | 100.00% |

## 5. Establish a Predictive Analysis Model

### 5.1. Pearson Correlation Analysis

First, the SPSS software is used to proceed with correlation analysis between the shape index of the road network in residential areas and the distance and directness coefficient. Pearson's simple correlation coefficient is used to measure the linear correlation, and the residential road network morphological index that affects the path connectivity is preliminarily confirmed. The smaller the directness and distance index are, the higher the convenience will be. Therefore, when the influence of each morphological factor coefficient is negative, its effect on connectivity is positive. The results of the analysis of the built environment indicators and the corresponding distance and directness coefficients of the residential areas in the three districts are shown in Table 9.

**Table 9.** Pearson correlation analysis.

| | PRD | | Path Distance | |
|---|---|---|---|---|
| | Pearson | Sig. | Pearson | Sig. |
| *N1* | −0.330 ** | 0.000 | −0.248 ** | 0.000 |
| *N2* | −0.051 | 0.466 | 0.647 ** | 0.000 |
| *N3* | 0.020 | 0.772 | 0.597 ** | 0.000 |
| *N4* | −0.101 | 0.151 | 0.098 | 0.165 |
| *N5* | −0.042 | 0.554 | 0.618 ** | 0.000 |
| *N6* | −0.057 | 0.416 | 0.597 ** | 0.000 |
| *N7* | −0.157 * | 0.025 | 0.434 ** | 0.000 |
| *N8* | −0.164 * | 0.019 | 0.415 ** | 0.000 |
| *N9* | −0.064 | 0.360 | −0.381 ** | 0.000 |
| *N10* | −0.189 ** | 0.007 | −0.406 ** | 0.000 |

**Table 9.** *Cont.*

|  | PRD | | Path Distance | |
|---|---|---|---|---|
|  | Pearson | Sig. | Pearson | Sig. |
| *N11* | −0.185 ** | 0.008 | 0.302 ** | 0.000 |
| *N12* | −0.089 | 0.207 | 0.422 ** | 0.000 |
| *N13* | −0.274 ** | 0.000 | −0.102 | 0.145 |
| *N14* | −0.139 * | 0.047 | 0.370 ** | 0.000 |
| *N15* | 0.011 | 0.880 | 0.481 ** | 0.000 |
| *N16* | −0.102 | 0.147 | −0.130 | 0.063 |

Note: ** significant at α = 0.01; * = significant at α = 0.05.

Pearson linear correlation analysis results show that the travel distance *D* is significantly correlated with the size of the residential area. The larger the residential area, the further the distance traveled by residents. The length of the long side of the residential area, the size of the area, and the total road length are all significantly correlated with the path distance (*D*), whereas the number of entrances and exits, intersection density, and the X ratio are significantly correlated with pedestrian route directness (*PRD*). Except for the "ratio" indicators, all the other variables have a great impact on the path distance, and the factors affecting the path directness are mainly the nodes and the street patterns.

### 5.2. Multiple Linear Regression Analysis to Establish a Comprehensive Model of Connectivity

Pearson correlation analysis can only help to decipher a single relationship between variables, and a regression model can determine which morphological elements of the road network exert a comprehensive influence on path directness and distance in the case of eliminating collinearity. Through the principal component and the dimensionality reduction analysis of 16 sets of explanatory variables representing the shape of the residential road network, the principal component factors with sufficient explanatory ability can be extracted, thereby eliminating multicollinearity between the explanatory variables. As shown in Table 10, principal component analysis yields four sets of elements.

**Table 10.** Factor load matrix (component matrix).

|  | Main Component | | | |
|---|---|---|---|---|
|  | *F1* | *F2* | *F3* | *F4* |
| *N8* | 0.950 | 0.233 | 0.065 | 0.019 |
| *N9* | 0.946 | 0.261 | 0.008 | −0.032 |
| *N6* | 0.938 | −0.065 | −0.086 | −0.029 |
| *N14* | 0.890 | 0.293 | −0.093 | −0.100 |
| *N12* | 0.867 | 0.192 | −0.308 | −0.029 |
| *N5* | 0.866 | −0.371 | −0.104 | −0.029 |
| *N11* | 0.811 | 0.332 | 0.310 | −0.050 |
| *N2* | 0.799 | −0.310 | −0.211 | 0.406 |
| *N3* | 0.780 | −0.457 | −0.028 | −0.319 |
| *N15* | 0.674 | −0.243 | 0.518 | 0.121 |
| *N10* | −0.037 | 0.905 | −0.007 | −0.070 |
| *N7* | −0.160 | 0.852 | 0.069 | −0.066 |
| *N1* | 0.029 | 0.477 | 0.154 | 0.191 |
| *N13* | 0.185 | 0.299 | 0.768 | 0.047 |
| *N16* | 0.134 | 0.565 | 0.624 | −0.188 |
| *N4* | 0.062 | 0.182 | −0.184 | 0.947 |

Principal component *F1*—area size of the residential area and road network complexity: number of nodes, number of links, total road length, cell roads, T-junctions, area size, X-junctions, length of the long side, length of the short side, and culs-de-sac. These 10 variables are of a high load on the component factor *F1*, which means that they are highly

correlated with the first factor. The component *F1* is explained by "the sale of residential area and road network complexity".

Principal component *F2*—density of residential road network elements. The principal component *F2* is most closely related to the three indicators of intersection density, road network density, and number of entrances and exits. The first two are indicators of density, and the common feature of these three indicators is that in different residential area samples, the larger the indicator value, the denser the distribution of factors. Therefore, the principal component *F2* is explained by the "density of residential road network elements".

Principal component *F3*—residential road network connectivity. The highest load on the principal component *F3* is the X ratio and the cell ratio. Both of these variables reflect the proportion of the elements of a high-connectivity road network. Generally speaking, the larger the value, the better the connectivity of the network and the smoother the traffic. Thus, the principal component *F3* is explained by the "residential road network connectivity".

Principal component *F4*—boundary shape of the residential area. The last principal component *F4* is only the highest loaded on the ratio of the long and short sides, so it is explained by the "boundary shape of the residential area".

### 5.2.1. Establish Equation of Path Distance

According to the regression equation coefficients provided in the multivariate linear regression model analysis of the path distance (Table 11), the multiple linear regression equation can be obtained:

$$D = K1 + 101.176F1 - 65.734F2 + 22.025F4 - 15.742F3 \tag{6}$$

**Table 11.** Coefficient in the final model of the path distance equation.

| Model | Standardized Coefficients | | t | Sig. | Collinear Statistics | |
|---|---|---|---|---|---|---|
| | Standard Error | Beta | | | Permission | VIF |
| (constant) | 7.869 | | 54.357 | 0.000 | | |
| Principal component *F1* | 7.890 | 0.617 | 12.824 | 0.000 ** | 1.000 | 1.000 |
| Principal component *F2* | 7.890 | −0.401 | −8.332 | 0.000 ** | 1.000 | 1.000 |
| Principal component *F4* | 7.890 | 0.134 | 2.792 | 0.006 ** | 1.000 | 1.000 |
| Principal component *F3* | 7.890 | −0.096 | −1.995 | 0.047 * | 1.000 | 1.000 |

Dependent variable: path distance. Note: $N = 204$; ** significant at $\alpha = 0.01$; * = significant at $\alpha = 0.05$.

In the formula, the path distance is in meters, and the other principal components are normalized values. In the formula, *K1* is a constant 427.733; the principal component *F1* is the area size of the residential area and road network complexity; the principal component *F2* is the density of the residential road network; the principal component *F3* is the residential road network connectivity; and the principal component *F4* is boundary shape of the residential area.

### 5.2.2. Establish Equation of Pedestrian Route Directness

According to the regression equation coefficients provided in the multivariate linear regression model analysis of *PRD* (Table 12), multiple linear regression equations can be obtained:

$$PRD = K2 - 0.095F2 - 0.067F3 \tag{7}$$

The *PRD* has no unit, and the other principal components are standardized values. In the formula, *K2* is a constant 1.744; the principal component *F2* is the density of residential road network elements; and the principal component *F3* is the residential road network connectivity. From multiple regression analysis, it can be found that:

- The area size of the residential area and road network complexity have the greatest impact on the path distance, and they are positively correlated. The larger the scale is, and the more plentiful the internal elements are, the larger the egression distance will be;

- The density of the residential road network elements affects the path distance secondarily and is negatively correlated. The denser the distribution (increase in the intersection density, road network density, and the number of entrances and exits), the shorter the travel distance will be, and the accessibility can be improved;
- The boundary shape of the residential area is positively correlated with the path distance, indicating that the larger the ratio of long and short sides, the closer the shape of the residential area is to the long strip, and the longer the travel distance; the connectivity of the residential road network is negatively correlated with the path distance, and the better the connectivity (the higher the X ratio, the higher the cell ratio), the shorter the travel distance.

**Table 12.** Coefficient in the final model of the *PRD* equation.

| Model | Standardized Coefficients | | t | Sig. | Collinear Statistics | |
|---|---|---|---|---|---|---|
| | Standard Error | Beta | | | Permission | B |
| (constant) | 0.028 | | 62.258 | 0.000 | | |
| Principal component *F2* | 0.028 | −0.237 | −3.390 | 0.001 ** | 1.000 | 1.000 |
| Principal component *F3* | 0.028 | −0.167 | −2.399 | 0.017 * | 1.000 | 1.000 |

Dependent variable: *PRD*. Note: *n* = 204; ** significant at α = 0.01; * = significant at α = 0.05.

*5.3. Binary Logistic Regression Analysis and Connectivity Judgement Model*

Further mathematical modeling using the binary logistic regression analysis method is built to establish an evaluation equation for the road network connectivity. Using this equation, the road network samples in residential area can be distinguished with "acceptable" and "unacceptable" connectivity. Therefore, this can provide a more accurate basis for judging whether the network connectivity can be accepted.

Use the path distance (*D*), which less than 400 m, and the *PRD*, which less than 1.5 as the standard, and four states are assumed for the "connectivity" indicator: (1) *D* less than 400 m, *PRD* less than 1.5 (acceptable); (2) *D* greater than 400 m, *PRD* less than 1.5 (acceptable); (3) *D* less than 400 m, *PRD* greater than 1.5 (acceptable); (4) *D* is greater than 400 m, *PRD* greater than 1.5 (unacceptable). The first, second, and third categories are defined as "acceptable" path patterns, and the fourth category is classified as an "unacceptable" path pattern. Whether the connectivity is "acceptable" is predicted by inputting the indicators of the road network form in residential areas.

The results of binary logistic regression analysis show that there is a mathematical equation correlation between the road network form in residential areas and the connectivity, and its regression equation is: *Logit P* refers to type of the road network:

$$Logit\ P = 4.051 + 0.380N1 - 0.015N2 - 0.021N3 + 39.159N5 + 0.006N10 - 3.025N13 - 0.152N15 + e \tag{8}$$

$$Logit\ P = ln\frac{P}{1-P} \tag{9}$$

where *P* represents the probability when the explanatory variable takes a value of 1, which is the probability when the connectivity is acceptable. With this set of equations, the connectivity level prediction of the residential area sample can be performed. A value of *P* less than 0.5 indicates that the connectivity of the road networks in the residential area may not be accepted, and a parameter greater than 0.5 may be accepted. The equation shows that the level of connectivity is positively correlated with the number of entrances and exits, the area size, the intersection density, and the X ratio, and it is negatively correlated with the length of the long side, the length of the short side, and the culs-de-sac. In the formula, the equations of different road network types are parallel to each other, and the model is adjusted by the difference value (e) of the reference group (outer cell road).

Further, the 204 samples were distinguished by "acceptable" and "unacceptable" connectivity, and the classification results were compared with the results of the regression equation simulation analysis. The residential area samples with a connectivity level

consistent with the simulation results have a value for reference (acceptable) and optimization (unacceptable). The path distance and *PRD* of the 204 residential areas are directly measured and tested, and the correct rate of the prediction model is about 80%.

Some Chinese scholars have studied the relationship between transport facilities and carbon emissions and found that road network optimization measures, including increasing road intersections and the number of travel routes, can effectively reduce the carbon emissions of daily traffic trips. At the same time, the optimization of the road network shape can increase the residents' travel mode mix and travel convenience, reduce the willingness to use motor vehicles to a certain extent, and improve the willingness to walk [42,43]. The carbon peak situation of the urban residential areas in the future is critical to China's carbon peak and carbon neutrality commitment [44]. The above evaluation model for road network connectivity in residential areas in this paper better quantifies the impact factors of more specific road network forms and their impact on road connectivity. By establishing this connectivity judgement model, we can guide the optimization of road networks in residential areas in application. Meanwhile, the improvement of the density, accessibility, and diversity of the road network in residential areas can contribute to a reduction in travel carbon emissions.

## 6. Conclusions and Suggestions for Improvement

### 6.1. Conclusions

The planning and design of urban residential environments are important for reducing carbon emissions in China [45,46]. This paper is carried out at the residential area and block level to study the internal road network system of residential areas, the road traffic system in the district, and the connection between the two. The indicators of the built environment of residential areas are quantitatively measured, the path distance (*D*) and pedestrian route directness (*PRD*) methods are used to measure and calculate the road connectivity, and the relationship between the two is discussed. The following conclusions are drawn.

- Although road network density inside the residential area is high, the status quos generated by the closed residential area mode, such as the large scale of residential area, few entrances and exits, low road connectivity, low connectivity between internal roads and urban roads, and internal roads unable to participate in urban road traffic, are triggering traffic congestions in urban roads.
- The path distance *D* is significantly related to the size of the residential area. The larger the residential area, the further the travel distance. More than 60% of the residential areas in the three districts mentioned above in Nanjing have a travel distance of more than 450 m, making it difficult for residents to walk.
- The directness levels of the residential road network in the three districts are all unsatisfactory, and the average value is greater than 1.5. More than 50% of the residential areas in these three districts have a *PRD* greater than 1.6, indicating the presence of significant travel detours.
- Pearson linear correlation analysis shows that the length of the long side, the area size, and the total road length are significantly correlated with the path distance (*D*) index, whereas the number of entrances and exits, the intersection density, and the X ratio are significantly correlated with pedestrian route directness (*PRD*).
- The results of multiple regression analysis show that the area size of the residential area and road network complexity have the greatest impact on the path distance, and they are positively correlated. The larger the scale, and the more plentiful the internal elements, the larger the egression distance will be; the density of residential road network elements affects the path distance secondarily, and it is negatively correlated.
- The denser the distribution (increase in the intersection density, road network density, and the number of entrances and exits), the shorter the travel distance will be, and the accessibility can be improved.

- The boundary shape of the residential area is positively correlated with the path distance, indicating that the larger the ratio of long and short sides, the closer the shape of the residential area is to the long strip, and the longer the travel distance.
- The connectivity of the residential road network is negatively correlated with the path distance, and the better the connectivity (the higher the X ratio, the higher the cell ratio), the shorter the travel distance.
- The results of binary logistic regression analysis show that the level of connectivity is positively correlated with the number of entrances and exits, the area size, the intersection density, and the X ratio, and it is negatively correlated with the length of the long side, the length of the short side, and the culs-de-sac. The path distance and *PRD* of 204 residential areas are directly measured and tested, and the correct rate of the prediction model is about 80%.

### 6.2. Applications

In this paper, the mathematical model of road network form and connectivity in residential areas is established through data investigation and quantitative analysis, and the sample optimization is tested. On this basis, it summarizes the methods that can effectively improve the "acceptable" level of connectivity of the road network in residential areas, and it can be used to guide planning and design in the following aspects:

- First, in terms of openings and borders, the bypass distance can be effectively reduced by increasing the number of entrances and exits, and the utilization efficiency of the road network in residential areas can be improved. The result of multiple linear regression shows that the square residential area is more conducive to shortening the travel distance than the long strip residential area. Therefore, in the planning process, the boundary length and proportion of the residential area should be strictly controlled to avoid using the narrow and long marginal plots as residential land and reduce the practice of incorporating the urban landscape belt into the residential side.
- Second, in terms of scale and density, the planning department should properly control the size of the residential area within a certain range and require designers to ensure the connectivity level of the road network within the residential area, so as to effectively reduce the travel distance and promote the occurrence of walking behavior. When the area of the residential area is limited, increasing the number of intersections can create more path choices, and giving priority to increasing X intersections has a more obvious effect on improving connectivity. Under the condition that the road form is limited, appropriately reducing the residential area can also achieve the effect of improving the intersection density, but it should ensure that the overall traffic and functional layout are not affected.
- Third, in terms of morphology, the results of correlation analysis and mathematical modeling show that the road network connectivity of a grid shape is better than that of a tree shape. During the design process, connectivity can be improved by appropriately increasing the X-shape ratio. The most common method is to increase the connectivity of the road network by increasing the connection of a parallel road network through branch roads or converting some T-intersections into X-intersections. On the other hand, in the design process of ring roads and culs-de-sac, the number of culs-de-sac is generally controlled to improve connectivity by reducing culs-de-sac or making the culs-de-sac into open loops. However, it is also found in the research that the culs-de-sac have the function of improving privacy in the design of residential areas. Therefore, in the process of design optimization, the roads in front of residential buildings should not be reduced blindly, but the culs-de-sac that are discontinuous due to road layout defects or occupy unnecessary land should be optimized.

*6.3. Limitations*

In terms of evaluation indicators, this paper adopts the objective connectivity level as a measure of the convenience of the residential road network, but ignores the difference among different residents (age, gender, etc.) in the subjective point of view of the convenience degree of travel. Even for the same person, feelings on the path to different destinations may also be different. It is necessary to add the quantified values of the subjective feelings of the residents to the data correction in the study, or to include the controlling variables of the resident attributes in the mathematical analysis, and the variables adopted in this paper are all objective morphological indicators where errors will certainly exist.

In the aspect of path analysis, the path from the central point of the residential area to the intersection of adjacent urban roads represents the travel route of residents in a certain residential area. Although this method can be used in different residential areas, it also ignores the impact exerted by the surrounding urban facilities on the residents. The actual design of the residential area should not only consider the layout of surrounding commercial, entertainment, and cultural facilities, but it should also make full use of the surrounding natural resources. These factors may affect the layout of buildings and road networks and the set of entrances and exits.

In mathematical analysis, although it takes a lot of time and effort to adjust the binary logistic regression equation to obtain the best-fitting result, it is inevitable that some of the explanatory variables will be removed in the modeling process in the early research to eliminate collinearity, such as the ratio of the long and short sides, total road length, road network density, etc. These variables are not reflected in the conclusions of the design.

*6.4. Further Works*

- The calculation method of the size of the residential area needs great improvement: "length of the long side * length of the short side = area of the residential area" is the maximum diagonal method of the residential area [47].
- In view of the reality in China, a distinction should be made between two types of travel modes when calculating entrances and exits: driving and walking. It is recommended to increase the density of the calculation of entrances and exits instead of a simple quantity index of entrances and exits, as well as to distinguish the density of entrances and exits for cars and pedestrians.
- Only the relevant indicators of the road network pattern of the built-environment indicators are measured, and the building capacity index of the residential area, such as the plot ratio, building density, and resident population, are not included. It is recommended to add this part of the calculation of the built-environment indicators.
- In the calculation of road network density, only the road network density in the residential area was calculated previously, and the road connectivity value of the block scale (1–3 km radius) was not calculated. It is recommended to increase the road network density calculation at the block level.

**Author Contributions:** Conceptualization, Y.Z. and R.W.; Data curation, Y.Z. and R.W.; Formal analysis, R.W. and G.C.; Funding acquisition, Y.Z.; Investigation, R.W., G.C. and X.W.; Methodology, Y.Z. and R.W.; Project administration, Y.Z.; Resources, Y.Z.; Software, R.W. and G.C.; Supervision, Y.Z.; Validation, Y.W.; Visualization, R.W. and Y.W.; Writing—original draft, R.W.; Writing—review and editing, Y.Z. and R.W. All authors have read and agreed to the published version of the manuscript.

**Funding:** This research was funded by the National Natural Science Foundation of China, grant number 51978142; the Soft Science Research Project of Ministry of Housing and Urban-Rural Development of China, grant number 2012-R1-7; and the China Scholarship Council, grant number 201706095001.

**Informed Consent Statement:** Informed consent was obtained from all subjects involved in the study.

**Data Availability Statement:** The data are proprietary or confidential in nature and may only be provided with restrictions. The data presented in this study are available on request from the corresponding author.

**Acknowledgments:** We thank Jiaxin Dong, Xianchao Tang, Li Ren, and Y.W. from the School of Architecture in Southeast University for their assistance in research, analysis, and writing.

**Conflicts of Interest:** The authors declare that the research was conducted in the absence of any commercial or financial relationships that could be construed as a potential conflict of interest.

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
