# Peer review of "The Influence of Built-Environment Factors on Connectivity of Road Networks in Residential Areas: A Study Based on 204 Samples in Nanjing, China"

_buildings, doi:10.3390/buildings13020301_

Round 1
Reviewer 1 Report
The article addresses connectivity index of road networks in China, investigating what factors/features of the residential areas affects it the most. The findings are relevant as they can support effective planning in the years to come.
The article is well written and clear. Methodology is consistent and sound, as well as limits are discussed. I recommend accepting the paper in its present form, with just very small revisions:
- Abstract could be reduced, especially in results description, by selecting most relevant findings.
- Sentence at lines 38-40 in unclear
- The aim and objective of the paper could be highlighted more before going into the methodology (before paragraph 3).
- Impact of the research findings could be discussed more in the conclusions, e.g., potentiality of the findings to be included in planning guidelines etc.
Reviewer 2 Report
The paper presents the results of a thorough analysis of residential areas in Nanjing, China to investigate the influence of building factors on the connectivity of road networks.
I have only a few comments:
· Please, clarify the aim of your research and what gap in the existing knowledge aims to cover, in the Introduction.
· The discussion of the results should become in the light of previous research and better highlight how greenhouse gas emissions could be reduced.
· What are your suggestions for policy-makers and urban planners locally and globally according to your analysis results? In the Conclusions section.
Reviewer 3 Report
It is a very good research, full of deep analysis flaws inside the development of the paper. It is well balanced and interesting for the reader. The balance of technical part and development of the idea is well maintained during all the deployment of the ideas. I think it can be accepted in the present form.
Author Response
Thank you for reading and commenting on this paper! In the revised version, we emphasized the application and guiding significance of the study, and revised some words.
Reviewer 4 Report
This is a good paper on the relationship between built environment and road networking in residential areas in Nanjing, China. The paper is well written and well structured. The state of the art is satisfactory. Methodology seems sound, and conclusions are in tune with the research findings. The paper could perhaps be enriched if, here and there, some emphasis was put on the importance of implementing mixed-mode uses, in the studied areas. In page 18, line 562 please replace "Person" by "Pearson" (linear correlation).
Author Response
Thank you for your reading and suggestions! In chapter 5.3, we added a description of carbon emission reduction by mixed-mode, and revised some words in the article.
Round 2
Reviewer 2 Report
The authors improved the manuscript satisfactorily.